# Quality of Life and Disease Impact of Atopic Dermatitis and Psoriasis on Children and Their Families

**DOI:** 10.3390/children6120133

**Published:** 2019-12-02

**Authors:** Chan Ho Na, Janice Chung, Eric L. Simpson

**Affiliations:** 1Department of Dermatology, College of Medicine, Chosun University, Gwangju 61453, Korea; chna@chosun.ac.kr; 2School of Medicine, Creighton University, Omaha, NE 68102, USA; JaniceChung@creighton.edu; 3Department of Dermatology, Oregon Health & Science University, Portland, OR 97239-4501, USA

**Keywords:** atopic dermatitis, psoriasis, quality of life, QoL tools, multidisciplinary approach, education, psychological support

## Abstract

Atopic dermatitis (AD) and psoriasis are common chronic skin diseases affecting children. These disorders negatively impact the quality of life (QoL) of patients in health-related aspects such as physical, psychosocial, and mental functioning. This health impact is more accurately represented when accounting for the numerous comorbidities associated with each disorder, and the impact the disorders have on patients’ families. A number of QoL tools have been developed and can be routinely implemented in the evaluation of QoL in pediatric patients and their caregivers. Ways to improve QoL include a multidisciplinary approach to care, education, and psychological support.

## 1. Introduction

Atopic dermatitis (AD) and psoriasis are common cutaneous diseases among children. AD is a chronic pruritic inflammatory skin disease that often presents as ill-defined, erythematous weeping or crusted, papules and/or plaques, and is frequently accompanied by a personal or family history of type I allergies, asthma, and allergic rhinitis [1]. It affects up to 17.5% of children worldwide, 11% of children in the US [2,3], and approximately one-third of children with AD have a moderate-to-severe disease [4]. A variety of factors, including skin barrier defects, reduction in antimicrobial peptides, dysfunctional innate immune response, and Th2-skewed skin and systemic inflammation are implicated in the pathogenesis of AD [5]. Itch is the main symptom of AD that can lead to frequent scratching, painful skin, loss of sleep, and skin infections [1,6].

Like AD, psoriasis carries a genetic predisposition and typically presents as well-demarcated, salmon-colored plaques with flaky, silvery-white scales. Differentiating between pediatric psoriasis and AD is usually straightforward clinically with no specific testing required (Table 1). Psoriasis is much less prevalent in children than AD with the estimated prevalence of childhood psoriasis varying from 0% to 1.37% globally. Psoriasis often first appears between the ages of 15 and 35, however, about a third of patients developed the condition during the first two decades, and 10–15% reported onset before age 10 [7,8,9]. It is a chronic relapsing inflammatory disease triggered by the expression of Th17 and TNF/IL-17 synergistic cytokines primarily [10]. Patients with psoriasis often describe their pruritus as a stinging or burning sensation, and this itch has been reported to be the most bothersome symptom in about 60–90% of patients [11,12].

In recent decades, the medical and scientific community is better recognizing the importance of assessing the quality of life (QoL) impact medical conditions have on patients. While QoL is a broad concept including standard of living, community, and family life, health-related QoL (HRQoL) is a multidimensional construct including emotional, physical, social, and functional well-being related to health, particularly within chronic diseases. An already large and growing body of literature concludes that both AD and psoriasis profoundly impact the HRQoL of affected children and their families, especially in more moderate to severe disease [13,14,15]. Thus, the assessment of HRQoL is very important in these diseases as it supplements the conventional clinical exam or scoring systems, captures the effects of the illness on patients and their families, and further measures whether a treatment is actually effective [16,17].

## 2. Measurement of QoL

Several studies reveal that provider-based assessments do not correlate well with patient-reported QoL assessments emphasizing the need for formal QoL assessments in the clinic to better understand the patient perspective of their disease. There are a variety of different instruments at the provider’s disposal to facilitate the measurement of QoL, including *dermatology-specific* and *disease-specific* QoL measures (Table 2). While *dermatology-specific* QoL scales are applicable in all cutaneous diseases and allow for comparisons across different skin diseases, *disease-specific* QoL instruments focus on specific patterns of a certain disease, which can provide much better insight into its particular characteristics, as well as direct ways to improve QoL [17,18]. These tools vary widely with respect to the age of target populations, domains assessed, and scoring algorithms.

QoL tools are usually classified as *self-reported* (for older children and adolescents) or *proxy-reported* (for infants and younger children) depending on the ages of the respondents. Cartoon versions (e.g., cartoon version of CDLQI [19]) for young children are becoming increasingly popular [17]. Regular use of these assessments in patients with chronic dermatitis has been recommended because not only can they be used to assess QoL and better understand the patient perspective and impact of the disease, but they can also be used to measure the beneficial effects of treatment [1,20,21,22,23].

In clinical trials of skin disease, both *dermatology-specific* QoL tools (such as the Dermatology Life Quality Index (DLQI), the Children’s Dermatology Life Quality Index (CDLQI), the Infants’ Dermatitis Quality of Life Index (IDQoL)) and *disease-specific* QoL tools (such as the Dermatitis Family Index (DFI), the Childhood Atopic Dermatitis Impact Scale (CADIS)) have been commonly used in measuring pediatric and family QoL [1,24,25,26,27]. Recently, the Harmonising Outcome Measures for Eczema (HOME) initiative identified QoL as an essential domain to be measured in all clinical trials in AD [28]. An April 2019 consensus conference by the HOME initiative identified and recommended that the following instruments should be included in all trials of AD: DLQI (adults), CDLQI (children), and IDQoL (infants) [29].

## 3. Disease Impact of AD

### 3.1. AD Impact Relative to Other Diseases

AD impacts childhood HRQoL to a greater extent than many other chronic cutaneous diseases including urticaria, alopecia, acne, and localized eczema. AD was found to be equivalent in impact to other non-dermatologic chronic childhood disorders and only second to cerebral palsy [14]. It has been suggested that raising a child with AD could be more impactful to the family than raising a child with type I diabetes [41].

Itch and sleep-related deficits are closely associated with QoL of AD. Children can become irritable and inattentive when experiencing severe pruritus, and parents often find it difficult to keep their child from scratching [42]. Additionally, pruritus is highly associated with sleep disturbance, which has been estimated to affect as much as 60% of children or 83% of children with flares [43]. The effects of such sleep disturbance include decreased sleep efficiency from waking up throughout the night, trouble getting to sleep, reduced total sleep time, difficulty waking up, as well as daytime drowsiness and irritability [44]. Sleep disturbance does not simply affect children with AD, but also their family members who might have to provide medications or reassurance during the night, or from being awakened while cosleeping [45].

### 3.2. Psychosocial/Mental Comorbidity

There has been increasing evidence of the elevated risk of developing psychosocial and mental comorbid diseases in childhood AD [46,47]. According to the available literature, psychological comorbidities were more prevalent in children with AD than those with leukemia or epilepsy [48]. Mental and emotional sequelae, including cognitive and functional impairment, as well as behavioral problems often extend far beyond the actual physical manifestations, especially in patients with severe disease [49,50,51]. Psychosocial abnormalities of children with AD appear in various ways. Behavioral and maladjustment problems, sibling rivalry, abnormal psychological development, childhood low self-esteem, and lack of socialization skills were all reported [41]. Children with AD also showed excessive dependency, clinginess, and fearfulness [52]. Damaged social functioning results in missing outdoor activity or avoiding outside work altogether [53]. Emotional and physical fatigue owing to chronic sleep loss may negatively affect social relationships and QoL of the adolescent AD patient, leading to an increased risk of depression and anxiety [54]. A cross-sectional study analyzing data from 92,642 US children reported that children with AD had increased risk of attention-deficit hyperactivity disorder (ADHD), depression, anxiety, conduct disorder, and autism [55]. Such disorders can continue or become exacerbated when the child’s AD remits [56].

Having AD appears to increase the risk of suicide and suicidal ideation. One study of Korean adolescents with AD found a significantly larger risk of suicide and suicidal ideation compared to adolescents without the disease, while another study from Korea showed a significant association between atopic dermatitis and suicidal behaviors for girls [57,58]. A meta-analysis confirmed this increased risk of suicidal ideation in children and adults with AD compared with healthy controls [59].

High levels of anxiety and depression have also been found in parents of a chronically ill child, especially mothers [60]. In particular, childhood AD negatively impacts maternal physical and mental wellbeing [61]. Parents often experience exhaustion, frustration, helplessness, feelings of guilt, sleep deprivation, and instability of spousal and other familial relationships [41,53]. They also have more absences from work, poor social activities, stress about child care, and challenges related to discipline than parents without chronically ill children [46]. When it comes to management of AD, many worries are related to disease triggers and medication use, including fear of using topical corticosteroids [62].

### 3.3. Impact of Comorbidities on QoL

Patients with moderate to severe AD have a greater risk of having comorbid asthma, hay fever, food allergies, cutaneous infections and possibly cardio-metabolic comorbidities, each of which may negatively impact QoL [63,64]. A number of concurrent atopic conditions (e.g., food allergy, allergic rhinitis, asthma) and severity of the child’s AD are related proportionally to the likelihood of psychological comorbidity [65]. Shorter stature, delayed growth, and early childhood obesity have been documented as possible physical comorbidities, particularly in children with severe AD [66,67,68].

## 4. Disease Impact of Psoriasis

### 4.1. Psoriasis Impact Relative to Other Diseases

Children with psoriasis are negatively impacted in their educational, physical, emotional, and social wellbeing in comparison to their healthy peers [69]. Their overall QoL impairments are significantly greater than those in children with epilepsy, enuresis, diabetes, and vitiligo [14,70]. The HRQoL of children with moderate-to-severe psoriasis is similar to that of their peers suffering from chronic disorders including arthritis and asthma, although pediatric psychiatric disorders cause the most impairment [15,71]. Children with psoriasis commonly experience itching, which causes considerable daily impairment and sleep disturbance, albeit to a lesser extent than that of adolescents with AD and adults with psoriasis [72,73,74,75].

### 4.2. Psychosocial/Mental Comorbidity

While there are a number of studies on the psychological and mental wellbeing of adults with psoriasis, there has not been much literature regarding children with psoriasis. A retrospective study of 61 pediatric psoriasis patients reported concurrent emotional stress (54%) and psychiatric morbidity (9.8%) [76]. Additionally, pediatric patients with psoriasis were shown to have an approximately 25% to 30% greater risk of developing psychiatric disorders such as depression and/or anxiety versus children without psoriasis [77]. Anxiety or depression may stem from experiences of embarrassment, shame, behavior avoidance, teasing, bullying, stigmatization, disrupted body image, decrease in self-confidence, and social isolation, resulting in some patients being prescribed psychotropic medications [77,78,79]. Psoriasis beginning in child or teenage years has been associated with more frequently developing flares in times of stress or trauma, and might be associated with higher disease severity and psychological comorbidities [80,81]. As many as 90% of children report psoriasis flares with stress, and having visible or severe lesions may cause further stress, potentially exacerbating the disease process itself in an escalating biopsychosocial cycle [81,82].

While mood disturbances such as feeling upset or sad as measured by parent-report in young children are severely impacted, adolescents find that appearance-related concerns, that is, ‘being on display,’ are more problematic [72,83]. Notably, adolescents with new-onset psoriasis may not be comfortable with their new appearance, which could lead to future impairment of HRQoL, such as the development of intimacy issues [84].

Although there are only a few studies for suicidality in children with psoriasis, several studies have reported that younger patients with psoriasis have a greater likelihood of experiencing suicidality than older patients [85]. Thus, it is essential for physicians to identify psoriatic patients of all ages who may be at risk for suicidal ideation and behavior.

Ninety percent of family members of adults with psoriasis responded that the patient affects their own QoL [86]. One study noted that parents of children with psoriasis are adversely impacted in their QoL in aspects such as emotional and personal wellbeing, functioning within the family and society, and life pursuits [15]. In the study, participating parents responded that their child’s psoriasis particularly caused a substantial impact on their emotional well-being as concerns about their child were a source of stress, sadness, frustration, and depressed mood.

### 4.3. Impact of Comorbidities on QoL

Children with psoriasis have an increased risk of concomitant obesity, hypertension, heart disease, diabetes, and autoimmune disorders such as rheumatoid arthritis and Crohn’s disease, all of which may subsequently negatively impact QoL [87,88]. Psoriasis and obesity are both socially stigmatizing. The stigma surrounding psoriasis may cause patients to become embarrassed and have low self-esteem, and poor coping mechanisms may lead to avoidance of social situations, eating in excess, forgoing activity, and developing or worsening obesity. Therefore, pediatric psoriasis and its increased risk of obesity can contribute to the exacerbation of one another and overall decrease patient QoL [89,90].

## 5. Strategies to Improve QoL

The routine use of proper QoL questionnaires can be an effective tool in early assessment and intervention of treatment effects. Despite the advent of many new QoL evaluation tools for chronic pediatric AD, most were poorly validated and generally unavailable for use in routine clinical practice. Further validation studies and continued development of new QoL scales focused on feasibility for clinical practice are necessary. No QoL instrument has been endorsed by the HOME group for use in clinical practice yet, but the DLQI, IDQoL, and CDLQI instruments have been endorsed for clinical trials and given limited items on these questionnaires, they may be suitable for clinical practice. The Skindex-16 or Skindex-mini, which investigate the symptomatic, emotional, and functional aspects of an individual’s skin disorder, may also be suitable for clinical practice with the latter having only three questions [91]. Furthermore, given the growing knowledge of suicidal ideation and behavior in these dermatologic diseases, there is a need to develop and apply feasible psychiatric screening tools for this population as a preventative measure in the near future.

A multidisciplinary approach, including communication and collaboration between primary care providers, dermatologists, psychiatrists, pediatric specialists, nurses, social workers, nutritionists, and support groups, is essential for long-term control of these disorders, especially when psychological support is needed [92,93,94,95]. The team should educate the patients and their families on their diseases, with focuses on the chronic relapsing nature and difficulties inherent to the disease, preventative treatment, and control of acute exacerbations, and the necessities of frequent monitoring and psychological support. So-called “therapeutic patient education” helps patients and their caregivers better understand their condition and its treatment, leading to improvement of QoL and adherence to therapy; therefore, it should be provided particularly to patients and families who have experienced treatment failures or feel a lack of social support [96]. Physicians should also refer to a mental health specialist when necessary, and explain possible comorbidities to the patient as part of the management plan [1].

In recent decades, eczema schools consisting of structured age-related group training programs regarding practical tips of management of AD such as medical information, skincare, and coping with psychosocial problems, have been effectively implemented throughout Germany. Given their success, these schools are now being adopted in several countries, including Japan, Denmark, and the US [97,98,99]. Italian researchers compared conventional treatment of AD to treatment with the addition of regular educational programs; supplementation with reinforced education led to decreased levels of anxiety in the family and fostered interaction among the children, parents, and physicians [95]. Dutch researchers testing multidisciplinary training programs for 15 children with psoriasis and their parents reported improved patient QoL, reduced pruritus and scratching, strengthened illness awareness, diminished impact on family life, and decreased disease severity [92].

The participation in support groups such as the National Eczema Association, National Psoriasis Foundation, and online relevant forums should also be recommended. Such groups can enhance the multidisciplinary approach as they offer opportunities to exchange experiences among attendants, thereby enlightening others regarding the complexity of the disease and helping other individuals who face similar problems [100]. In particular, online forums may be helpful because anonymity allows individuals to express their feelings freely while eliminating the stress of appearance [101].

## 6. Conclusions

Overall, chronic childhood diseases such as AD and psoriasis can have a detrimental impact on social, personal, and emotional perspectives of QoL of patients and their parents (Table 3). Providers should routinely ask patients about how their skin disease impacts their quality of life, specifically how the disease affects sleep, social relationships, mood, and school. This provides a better understanding of the patients and helps guide the shared-decision-making process. Validated instruments designed to assess QoL of patients provide an additional level of understanding of a patient’s condition and allows for a standardized longitudinal assessment of therapeutic effects. The multidisciplinary approach, including participation in support groups, may lead to an enhanced understanding of the disease and therefore may improve QoL. Additional development and evaluation of more practical QoL scales to explore treatment efficacy, as well as additional support strategies for pediatric patients and their families, are needed.

## Figures and Tables

**Table 1 children-06-00133-t001:** Clinical manifestations of atopic dermatitis (AD) and psoriasis.

	AD	Psoriasis
Lesion configuration	Erythematous scaly, crusted papules, lichenification, excoriation	Sharply demarcated, erythematous scaly plaques
Scale	Fine	Silvery-white, thick
Oozing/crust	Common	Rare
Distribution	Flexor surfaces (neck, antecubital and popliteal fossae), face, chest, wrist, back of hands and feet	Extensor surfaces of elbows and knees, scalp, sacral region, chest, face, abdomen
Symptoms	Pruritus	Stinging, burning
Photographic example	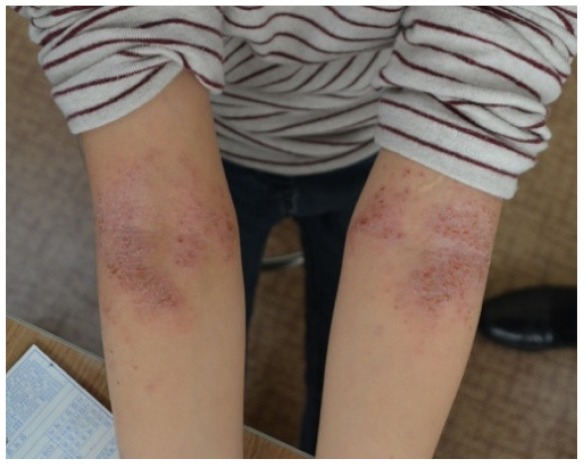	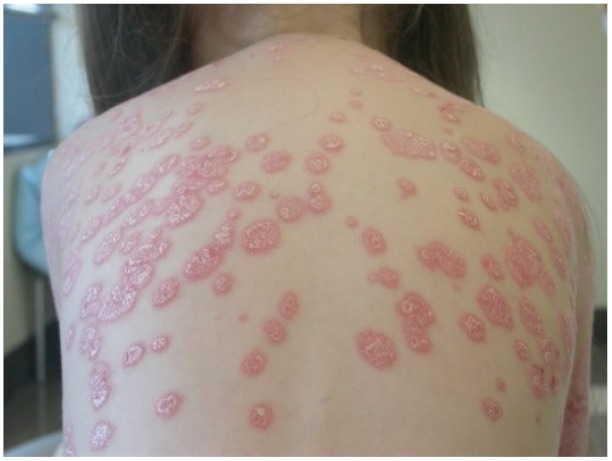

AD, atopic dermatitis.

**Table 2 children-06-00133-t002:** Instruments to assess health-related quality of life (HRQoL) of pediatric patients and their parents/caregivers.

Dermatology-Specific Tools	Disease-Specific Tools	
	AD	Psoriasis
- Dermatology Life Quality Index (DLQI) [30] - Children’s Dermatology Life Quality Index (CDLQI) [24] - Infants’ Dermatitis Quality of Life Index (IDQoL) [25] - Family Dermatology Life Quality Index (FDLQI) [31] - Skindex-Teen [32] - Toddler Quality of Life Survey [33]	- Dermatitis Family Index (DFI) [13] - Childhood Atopic Dermatitis Impact Scale (CADIS) [34] - Childhood Impact of Atopic Dermatitis (CIAD) [35] - DISABKIDS Atopic Dermatitis Module (DISABKIDS-ADM) [36] - Parents’ Index of Quality of Life in Atopic Dermatitis (PIQoL-AD) [37] - Quality of Life in Primary Caregivers of children with Atopic Dermatitis (QPCAD) [38] - The Quality of Life in Parents of Children with Atopic Dermatitis [39]	- Children’s Scalpdex in Psoriasis [40]

HRQoL, health-related quality of life; AD, atopic dermatitis.

**Table 3 children-06-00133-t003:** Disease impacts of AD and psoriasis in children.

	AD	Psoriasis
Overall QoL impairment	- Greatest negative impact on HRQoL among chronic skin disorders including urticaria, alopecia, acne, and localized eczema [14]	- Greater negative impact than those seen in children with epilepsy, enuresis, diabetes, and vitiligo [14,70] - Comparable impairment to those seen in children with other chronic diseases such as arthritis or asthma [15,71]
Itch and sleep disturbance	- Frequent scratching due to itching [42] - Decreased sleep efficiency, trouble getting to sleep, reduced total sleep time, difficulty waking up, daytime drowsiness, and irritability [43,44]	- Itching frequently [72,73,74] - Disturbed sleep due to pruritus and pain [75]
Psychosocial/mental comorbidities	- Behavioral and maladjustment problems, sibling rivalry, abnormal psychological development, childhood low self-esteem, and lack of socialization skills [41], excessive dependency, clinginess, fearfulness [52], and damaged social functioning [53] - Increased risk of ADHD, depression, anxiety, conduct disorder, and autism [55]	- Embarrassment, shame, behavior avoidance, teasing, bullying, stigmatization, disrupted body image, decrease in self-confidence, and social isolation [77,78,79] - Increased risk of anxiety and depression [77]
Comorbidities	- Comorbid asthma, hay fever, food allergies, cutaneous infections, and possibly cardio-metabolic comorbidities [63,64] - Shorter stature, delayed growth, and early childhood obesity [66,67,68]	- Increased risk of concomitant obesity, hypertension, heart disease, diabetes, and autoimmune disorders [87,88,89,90]
Impact for caregivers and family	- Greater impact than having a child with type 1 diabetes [41] - Sleep deprivation [45], anxiety, depression [60], exhaustion, frustration, helplessness, feelings of guilt, and instability of spousal and other familial relationships [41,53]	- Adverse impact on emotional well-being including stress, sadness, frustration, and depressed mood [15]

AD, atopic dermatitis; ADHD, attention-deficit hyperactivity disorder.

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
