# Peer review of "Quality of Life and Disease Impact of Atopic Dermatitis and Psoriasis on Children and Their Families"

_children, 2019, doi:10.3390/children6120133_

Round 1

Reviewer 1 Report

Chronic diseases consitute a serious clinical and social problem. mulfifaceted, complex and not entirely explored ethiopathogenisis is a stimulus for futher research. However, both the patient and the doctor expect effective treatment. It may be assumed that that holistic medicine should treat chronic diseases,including atopic dermatitis (AD) and psoriasis, in a model way.this is because these diseases pertain to all health aspects, and affect QoL of patients and their families.  the assessment and improvement of QoL is an indispensable component of proper clinical approach, especially in paediatric patients. The sumitted review paper on the effect of psoriasis and atopic dermatitis on the quality of life was to present how children"s health is affecetd and how AD and psoriasis affect patients"s families. In my opinion the Authors did not in fact write a review paper on the effect of AD and psoriasis on QoL. Listing standardised tools used for the assessment of QoL cannot be considered an aqequate approach toward the matter discussed. Instead of just listing the tools, the paper should describe their application and significance for the problem in question. the identification of the problem offers opportunities for its for its effective elimination. this  can be done e.g. using surveys for the assessment of QoL. The paper was to  present the effect of these particular illnesses on children QoL and on the patientst's families. I cannot find such information in the paper so I believe it is not sui table for publication.   Chronic diseases consitute a serious clinical and social problem. mulfifaceted, complex and not entirely explored ethiopathogenisis is a stimulus for futher research. However, both the patient and the doctor expect effective treatment. It may be assumed that that holistic medicine should treat chronic diseases,including atopic dermatitis (AD) and psoriasis, in a model way.this is because these diseases pertain to all health aspects, and affect QoL of patients and their families.  the assessment and improvement of QoL is an indispensable component of proper clinical approach, especially in paediatric patients. The sumitted review paper on the effect of psoriasis and atopic dermatitis on the quality of life was to present how children"s health is affecetd and how AD and psoriasis affect patients"s families. In my opinion the Authors did not in fact write a review paper on the effect of AD and psoriasis on QoL. Listing standardised tools used for the assessment of QoL cannot be considered an aqequate approach toward the matter discussed. Instead of just listing the tools, the paper should describe their application and significance for the problem in question. the identification of the problem offers opportunities for its for its effective elimination. this  can be done e.g. using surveys for the assessment of QoL. The paper was to  present the effect of these particular illnesses on children QoL and on the patientst's families. I cannot find such information in the paper so I believe it is not sui table for publication.   

Author Response

Point 1:

Chronic diseases consitute a serious clinical and social problem. mulfifaceted, complex and not entirely explored ethiopathogenisis is a stimulus for futher research. However, both the patient and the doctor expect effective treatment. It may be assumed that that holistic medicine should treat chronic diseases,including atopic dermatitis (AD) and psoriasis, in a model way.this is because these diseases pertain to all health aspects, and affect QoL of patients and their families.  the assessment and improvement of QoL is an indispensable component of proper clinical approach, especially in paediatric patients. The sumitted review paper on the effect of psoriasis and atopic dermatitis on the quality of life was to present how children"s health is affecetd and how AD and psoriasis affect patients"s families. In my opinion the Authors did not in fact write a review paper on the effect of AD and psoriasis on QoL.

 Response 1:

We are sorry that the reviewer did not feel that we discussed the impact of AD/psoriasis on QoL. In section 3 and 4, we reviewed 51 studies (Ref.14,15,41-89) describing how AD & psoriasis affect the QoL of patients and families. We believe this review was quite comprehensive and are not aware of any major papers left out of the discussion. Please provide more guidance on how to improve these sections regarding what specific papers to include and further points to make.

Point 2:

Listing standardised tools used for the assessment of QoL cannot be considered an aqequate approach toward the matter discussed. Instead of just listing the tools, the paper should describe their application and significance for the problem in question. the identification of the problem offers opportunities for its for its effective elimination. this  can be done e.g. using surveys for the assessment of QoL.

Response 2:

We defined QoL tools (section 2, lines 54-81) prior to describing QoL and disease impacts on children and their families (section 3,4, lines 82-183) to provide context for the necessity of such tools. Reviewer 2 and 3 actually requested more details on QoL tools, so we would prefer leaving this information in the paper and in this location. Our goal was to inform our readers about QoL impacts in AD and psoriasis by providing recent literature findings regarding QoL tools, disease impacts identified in QoL studies, and suggested methods to improve QoL. Accordingly, we are sure that such information in this paper might help provide the physician or the investigator a better understanding of QoL of children with AD and psoriasis and their families in the clinical setting.

Point 3:

The paper was to  present the effect of these particular illnesses on children QoL and on the patientst's families. I cannot find such information in the paper so I believe it is not sui table for publication.

Response 3:

Please refer to our response to point 1.

Reviewer 2 Report

The manuscript by Chanho Na aims to investigate

” Quality of life and disease impact of atopic dermatitis and psoriasis on children and their families.” While the study is interesting however, multiple serious concerns that limits the enthusiasm

Comments to the Author

Quality of life of atopic dermatitis and psoriasis on children is an interesting and important subject. It is worth being discussed and affirmed. Can more describe the “Methodology of Quality of Life Assessment” in the part of “Measurement of QoL”. Table 2. need to be more clearly presented. For example about the percentage.

Author Response

Point 1:

Quality of life of atopic dermatitis and psoriasis on children is an interesting and important subject. It is worth being discussed and affirmed.

 Response 1:

Thank you for your valuable comment. We agree.

Point 2:

Can more describe the “Methodology of Quality of Life Assessment” in the part of “Measurement of QoL”.

 Response 2:

Thanks for your feedback. We also think that it would be no exaggeration to say that the methodology of QoL assessment and methodology of psychometric properties is crucial. However, the request by the editors was to present the impact of AD/psoriasis on QoL. We feel a discussion on a very technical topic such as QoL methodology is beyond the scope of this paper. Groups such as COSMIN and HOME have recently reviewed the methodologies for the various QoL instruments of dermatology. Currently, however, no QoL tools for children with AD or psoriasis and their guidance can be highly recommended thus far, except for some tools (e.g., DLQI, CDLQI, and IDQoL) recommended by HOME. Therefore, we think that this review paper would be more meaningful by presenting known QoLs and solutions through studies using these tools, rather than introducing lots of QoL tools in a lengthy manner.

Point 3:

Table 2. need to be more clearly presented. For example about the percentage.

 Response 3:

The goal of Table 2 was to show the sheer number of options for measuring QoL in this population. It is not clear what the reviewer means by percentages as they are not presented in this table.

Reviewer 3 Report

The review summaries the disease burden of pediatric atopic dermatitis and psoriasis. Both of the inflammatory diseases do have severe impact over patients especially in childhood stage. The comorbidities and disease burden may raise the public and family awareness to take prevention or intervene the disease earlier.

Comments:

The review classified the QoL questionnaire into two categories, namely dermatological and disease-specific tools. The advantages and disadvantages of each QoL questionnaire are not explained. This information might help the physician or the investigator to have better understanding as to how to choose the questionnaire to obtain desired response. Since the review focus on children with atopic dermatitis or psoriasis. Does the article have done research in terms of the psychological burdens or the QoL questionnaire for the caretaker? This information might be incorporated. The article also mentioned about suicidal ideation in both atopic dermatitis and psoriasis. Are there any tools or survey that can help the family or the clinicians to detect the suicidal attempt earlier to prevent such tragedy?

Author Response

Point 1:

The review classified the QoL questionnaire into two categories, namely dermatological and disease-specific tools. The advantages and disadvantages of each QoL questionnaire are not explained. This information might help the physician or the investigator to have better understanding as to how to choose the questionnaire to obtain desired response.

 Response 1:

Thank you for your valuable feedback. We pointed out the pros and cons of dermatological and disease-specific tools in section 2 (lines 59-62):

‘While dermatology-specific QoL scales are applicable in all cutaneous diseases and allow for comparisons across different skin diseases, disease-specific QoL instruments focus on specific patterns of a certain disease, which can provide much better insight into its particular characteristics, as well as direct ways to improve QoL [17,18]’. We also agree with you that the choice of proper QoL scales is very important in clinical settings. As far as we know, however, there is no established gold standard with good feasibility and validity to assess QoL in these diseases, apart from some tools (e.g., DLQI, CDLQI, and IDQoL) recommended by HOME. Therefore, we think that an overall review regarding the advantages and disadvantages of each QoL instrument might be too lengthy and would not lead to a direct recommendation for proper QoL scales.

Point 2:

Since the review focus on children with atopic dermatitis or psoriasis. Does the article have done research in terms of the psychological burdens or the QoL questionnaire for the caretaker? This information might be incorporated.

 Response 2:

We also reviewed some studies regarding psychological burdens for the caretakers and described them in ‘psychosocial/mental comorbidities’ of section 3 and 4 as described below:

‘High levels of anxiety and depression have also been found in parents of a chronically ill child, especially mothers [60]. In particular, childhood AD negatively impacts maternal physical and mental wellbeing [61]. Parents often experience exhaustion, frustration, helplessness, feelings of guilt, sleep deprivation, and instability of spousal and other familial relationships [41,53]. They also have more absences from work, poor social activities, stress about child care, and challenges related to discipline than parents without chronically ill children [46]. When it comes to management of AD, many worries are related to disease triggers and medication use, including fear of using topical corticosteroids [62]’ (section 3, lines 121-128)

‘Ninety percent of family members of adults with psoriasis responded that the patient affects their own QoL [85]. One study noted that parents of children with psoriasis are adversely impacted in their QoL in aspects such as emotional and personal wellbeing, functioning within the family and society, and life pursuits [15]. In the study, participating parents responded that their child’s psoriasis particularly caused a substantial impact on their emotional well-being as concerns about their child were a source of stress, sadness, frustration, and depressed mood’ (section 4, lines 169-174).

Point 3:

The article also mentioned about suicidal ideation in both atopic dermatitis and psoriasis. Are there any tools or survey that can help the family or the clinicians to detect the suicidal attempt earlier to prevent such tragedy?

 Response 3:

Thanks for your good comment. We also think it is necessary to have some psychiatric tools for screening of suicidal ideation and behavior as a preventative measure ultimately. However, there have been only a few studies (questionnaire surveys) investigating the prevalence and risk factors associated with development of suicidal ideation and behavior for children with AD or psoriasis. We could not find any study introducing effective suicidal-related screening tools in pediatric patients with AD or psoriasis in dermatology, even in adult patients, despite the many scales available for general psychiatry such as the Suicide Ideation Scale (SIS), Suicidal Ideation Screening Questionnaire (SIS-Q), Suicidal Affect Behavior Cognition Scale (SABCS), etc. A recent systematic review of measurement scales regarding suicidal ideation and attitudes identified 29 different research scales for assessing suicidal ideation, and concluded that there is currently no gold standard (Ghasemi P, Shaghaghi A, Allahverdipour H. Measurement scales of suicidal ideation and attitudes: a systematic review article. Health promotion perspectives. 2015;5(3):156.). Given the growing knowledge of suicidal ideation and behavior in these diseases, however, we added a sentence describing needs for a development of such tools in the future as follows;

‘Furthermore, given the growing knowledge of suicidal ideation and behavior in these dermatologic diseases, there is a need to develop and apply feasible psychiatric screening tools for this population as a preventative measure in the near future’ (section 5, lines 194-196).

Round 2

Reviewer 3 Report

This is a revised version of the manuscript. The authors have replied properly.

Since this manuscript combine two diseases (atopic dermatitis and psoriasis) together, it might be more informative if the authors could also compare the different impacts between the diseases in a Table. Another thing is that the authors mentioned itching is more frequent in children psoriatic patients compared to adults with psoriasis.  Please discuss whether this is true in atopic dermatitis.   

Author Response

This is a revised version of the manuscript. The authors have replied properly.

Point 1.

Since this manuscript combine two diseases (atopic dermatitis and psoriasis) together, it might be more informative if the authors could also compare the different impacts between the diseases in a Table.

Resonse 1.

Thank you for this comment. We created table 3 comparing the disease impacts of both diseases and added it to section 6, line 231.

Point 2.

Another thing is that the authors mentioned itching is more frequent in children psoriatic patients compared to adults with psoriasis.  Please discuss whether this is true in atopic dermatitis.

Response 2.

Actually, we discussed that itching is less frequent in child psoriatic patients compared to adults with psoriasis, as in the lines 143-145 as copied below:

“Children with psoriasis commonly experience itching, which causes a considerable daily impairment and sleep disturbance, albeit to a lesser extent than that of adolescents with AD and adults with psoriasis [72-75].”

Every AD patient shows rashes with itch, as itch is the cardinal symptom associated with the diagnostic major criteria of AD regardless of age. We could not find any study regarding the difference in frequency of itch depending on age. Rather, there are some studies that suggest that severe itching tends to be correlated with older age in AD, which we think does not need to be mentioned in this article regarding children.